# Performing the ABC Method Twice for Gastric Cancer Risk Stratification: A Retrospective Study Based on Data from a Large-Scale Screening Facility

**DOI:** 10.3390/diagnostics13071284

**Published:** 2023-03-28

**Authors:** Satoru Mizutani, Yu Takahashi, Takeshi Shimamoto, Hideki Nakagawa, Hiroyuki Hisada, Kaori Oshio, Dai Kubota, Hiroya Mizutani, Daisuke Ohki, Yoshiki Sakaguchi, Seiichi Yakabi, Keiko Niimi, Naomi Kakushima, Yosuke Tsuji, Ryoichi Wada, Nobutake Yamamichi, Mitsuhiro Fujishiro

**Affiliations:** 1Department of Gastroenterology, Graduate School of Medicine, The University of Tokyo, 7-3-1 Hongo, Bunkyo-ku, Tokyo 113-8655, Japan; 2Kameda Medical Center Makuhari CD 2F, 1-3 Nakase, Mihama-ku, Chiba-City, Chiba 261-8501, Japan; 3Next-Generation Endoscopic Computer Vision, Graduate School of Medicine, The University of Tokyo, 7-3-1 Hongo, Bunkyo-ku, Tokyo 113-8655, Japan; 4Infection Control and Prevention Service, Graduate School of Medicine, The University of Tokyo, 7-3-1 Hongo, Bunkyo-ku, Tokyo 113-8655, Japan; 5Center for International Preventive Medicine, Graduate School of Medicine, The University of Tokyo, 7-3-1 Hongo, Bunkyo-ku, Tokyo 113-8655, Japan; 6Center for Epidemiology and Preventive Medicine, Graduate School of Medicine, The University of Tokyo, 7-3-1 Hongo, Bunkyo-ku, Tokyo 113-8655, Japan; 7Department of Endoscopy and Endoscopic Surgery, Graduate School of Medicine, The University of Tokyo, 7-3-1 Hongo, Bunkyo-ku, Tokyo 113-8655, Japan

**Keywords:** ABC method, gastric cancer, *H. pylori*, pepsinogen, risk stratification

## Abstract

The ABC method is a classification method used for stratifying the risk of gastric cancer. However, whether the ABC method should be performed only once or multiple times throughout an individual’s lifetime remains unclear. Therefore, this study aimed to analyze whether performing ABC screening twice in a lifetime is useful. We retrospectively analyzed the data of individuals who participated in health checkups in 2010 and 2015. We collected data on patient characteristics, pepsinogen levels, anti-*Helicobacter pylori* antibody titers, and the presence of gastric cancer. Overall, 7129 participants without a history of *H. pylori* eradication were included in this study. The participants’ average age in 2010 was 48.4 ± 8.3 years, and 58.1% were male. In addition, 11 and 20 cases of new *H. pylori* infection (0.15%) and spontaneous eradication (0.28%), respectively, were recorded. No significant difference was found in the incidence of gastric cancer between participants who underwent the ABC method once and those who underwent it twice (Group A: 0.16% vs. 0.16%; Group B: 0.47% vs. 0.39%; and Group C + D: 1.97% vs. 1.82%). Therefore, performing the ABC method twice, 5 years apart, does not significantly improve gastric cancer risk stratification.

## 1. Introduction

*Helicobacter pylori* (*H. pylori*) is a bacterial pathogen that infects more than half of the world’s population [1]. It is believed to be transmitted from person to person through the oral–oral, gastro–oral, or fecal–oral routes, frequently during early childhood [2,3]. *H. pylori* infection is associated with chronic gastritis, peptic ulcer, gastric mucosa-associated lymphoid tissue lymphomas, and gastric cancer.

The incidence of gastric cancer varies according to regions, with particularly high incidence rates in East Asia [4,5,6,7]. Upper gastrointestinal barium radiography and endoscopic gastric cancer screening are effective screening tests for gastric cancer. These tests allow for the early detection and treatment of gastric cancer, which can reduce mortality rates among patients [8]. Although upper gastrointestinal barium radiography and endoscopic gastric cancer screening are effective for detecting gastric cancer, they can be uncomfortable for patients and are associated with a small but significant risk of adverse events. In addition, the cost of these screening tests and the limited availability of endoscopists make it nearly impossible to screen a considerable proportion of the general population. Given these limitations, these tests should be reserved for high-risk cases. Therefore, effective methods for identifying high-risk groups, such as the measurements of pepsinogen (PG) level, anti-*H. pylori* antibody titer, and gastrin-17 level, are needed [9,10]. The ABC method, which combines the measurement of serum PG level and anti-*H. pylori* antibody titer, is an effective and convenient method of stratifying the risk for gastric cancer [11,12,13,14].

It has been reported that the ABC method is still useful for stratifying the risk of gastric cancer in Japan, even recently, where decreasing *H. pylori* infection rates have been reported [15]. Since new *H. pylori* infections in adults are rare, the ABC method is generally sufficient to be performed once in an individual’s lifetime. However, there is no clear evidence whether the ABC method is sufficient once in a lifetime. As false-negative and false-positive results may be obtained and gastric mucosal atrophy may progress over time, some facilities perform the ABC method yearly.

High rates of reinfection after *H. pylori* eradication therapy have been reported in areas with high rates of *H. pylori* infection; however, the occurrence of new *H. pylori* infections in adults without a history of *H. pylori* eradication is unclear [16]. Recently, it was reported that *H. pylori* antibody titer and PG level change significantly in the early period after *H. pylori* eradication, albeit slowly [17]. However, how much atrophic gastritis has progressed over time in non-infected or currently infected patients in recent years is still unclear. In addition, no recent data exist on whether the measurement of anti-*H. pylori* antibody titer or PG level at different time points can affect the results of the ABC method.

Therefore, this study aimed to determine whether performing the ABC method twice is useful for gastric cancer risk stratification and to investigate the rates of new *H. pylori* infections and spontaneous *H. pylori* eradication in Japan using recent large-scale health checkup data.

## 2. Materials and Methods

### 2.1. Ethics Statements

The protocol for this study was approved by the Ethics Committees of the University of Tokyo (2865-(3)) and registered with UMIN-CTR (UMIN000013761). Written informed consent was obtained from each participant, and this study was performed in accordance with the principles of the Declaration of Helsinki.

### 2.2. Study Design and Participants

This was a retrospective cohort study conducted using the data of individuals who underwent health checkups at the Kameda Medical Center Makuhari (Chiba-shi, Chiba, Japan) in 2010 and 2015. Individuals who underwent the measurement of serum *H. pylori* antibody titer and PG I and II levels were included. However, individuals with a history of *H. pylori* eradication or those who did not undergo endoscopy or upper gastrointestinal barium radiography in 2010 or 2015 were excluded. Esophagogastroduodenoscopy (EGD) was recommended if abnormal findings were observed on upper gastrointestinal barium radiography. Data on age; sex; body mass index (BMI); anti-*H. pylori* antibody titer; pepsinogen level; smoking habits; use of proton pump inhibitors (PPIs), anticoagulants, non-steroidal anti-inflammatory drugs (NSAIDs), or steroids; history of upper gastrointestinal tract surgery; the presence of gastric cancer; and history of *H. pylori* eradication in 2010 and 2015 were collected. Information on the history of *H. pylori* eradication, history of upper gastrointestinal tract surgery, smoking habits, and regular use of medications was confirmed using a questionnaire. Responses to the question on smoking habits were categorized into three, as follows: current, past, and never. Endoscopic data recorded up to December 2019 were referenced for determining the presence of gastric cancer.

### 2.3. Measurement of Anti-H. pylori Antibody Titer

The E-plate Eiken anti-*H. pylori* antibody kit (Eiken Chemical, Tokyo, Japan) was used the measure serum anti-*H. pylori* antibody titer in 2010, whereas the E-plate Eiken anti-*H. pylori* antibody II kit (Eiken Chemical Co., Ltd., Tokyo, Japan) was used in 2015. Although the manufacturer’s instructions state that the cut-off value for anti-*H. pylori* antibody titer is 10 U/mL, cases of *H. pylori* infection in patients with antibody titers of 3–10 U/mL are common [18,19]. Since April 2017, the cut-off value for the ABC method in Japan was adjusted to 3 U/mL to improve sensitivity [20]. As in previous studies, the cut-off value for negative *H. pylori* infection was set at <3 U/mL in this study. *Anti-H. pylori* antibody titers ranging from 3 U/mL to 9.9 U/mL were defined as “high-negative titers,” whereas values ≥ 10 U/mL were considered positive. Cases in the high-negative titer group included current *H. pylori* infections and post-eradication cases. To reduce the incidence of false-positive results, new infections and spontaneous eradication of *H. pylori* were defined as cases where the patient’s anti-*H. pylori* antibody titer changed from negative (<3 U/mL) to positive (≥10 U/mL), and those where the antibody titer changed from positive (≥10 U/mL) to negative (<3 U/mL) without *H. pylori* eradication therapy, respectively. The rates of new *H. pylori* infection and spontaneous *H.pylori* eradication were examined over the 5-year study period.

### 2.4. The ABC Method

LZ test EIKEN PG I and PG II (Eiken Chemical Co., Ltd.) were used to measure serum PG I and II levels. A PG test is considered positive if the PG I/II ratio and PG I are <3.0 and <70 ng/mL, respectively. In the ABC method, patients are categorized into four groups based on their PG and anti-*H. pylori* antibody test results, as follows: Group A, PG (−) and anti-*H. pylori* antibody (−); Group B, PG (−) and anti-*H. pylori* antibody (+); Group C, PG (+) and anti-*H. pylori* antibody (+); and Group D, PG (+) and anti-*H. pylori* antibody (−). We compared the risk of gastric cancer between participants classified using the ABC method in 2010 and those classified in both 2010 and 2015. Performing the ABC method twice was defined as assigning a participant to the same ABC group in 2010 and 2015, or to a higher group if the result obtained in 2015 was different (e.g., if the participant was assigned to Group B in 2010 and Group C in 2015, the participant’s group would be Group C). The number of participants in Group D was small; therefore, it was combined with Group C for the analysis of the incidence rates of gastric cancer.

### 2.5. Statistical Analysis

All statistical analyses were performed using JMP 16.0 (SAS Institute, Cary, NC, USA) and Python 3.9 (Python Software Foundation, Wilmington, DE, USA). Categorical and continuous variables were analyzed using Pearson’s chi-square or Fisher’s exact tests and student’s t-test, respectively. A two-sided *p*-value < 0.05 was considered statistically significant. Additional bootstrap tests with 1000 replications were performed to test for differences in the incidence rates of gastric cancer.

## 3. Results

A total of 9414 adults met the inclusion criteria. Of these, 2139 participants were excluded because they had a history of *H. pylori* eradication therapy. Of the remaining 7275 participants without a history of *H. pylori* eradication, 146 participants who had not undergone EGD or upper gastrointestinal barium radiography were excluded. Therefore, the remaining 7129 participants without a history of *H. pylori* eradication were analyzed. Subsequently, 6750 participants were analyzed after excluding cases of new *H. pylori* infections, spontaneous eradication, history of upper gastrointestinal tract surgery, and PPI use (Figure 1).

The clinical characteristics of the participants are shown in Table 1. The participants’ average age in 2010 was 48.4 ± 8.3 years, and 4145 (58.1%) were male (Table 1).

In 2010, 5641 (79.1%) participants tested negative for *H. pylori*, 338 (4.7%) had a high negative anti-*H. pylori* antibody titer, and 1150 (16.1%) tested positive for *H. pylori*. However, in 2015, 5705 (80.0%) participants tested negative for *H. pylori*, 331 (4.6%) had a high negative anti-*H. pylori* antibody titer, and 1093 (15.3%) tested positive for *H. pylori*. Overall, 11 and 20 cases of new infections (11/7129 (0.15%)) and spontaneous eradication (20/7129 (0.28%)), respectively, were recorded (Table 2).

Sex, age, BMI, history of upper gastrointestinal tract surgery, smoking, and use of PPIs, anticoagulants, steroids, or NSAIDs were analyzed to determine the odds ratios for the new *H. pylori* infection. For the new infections, the non-adjusted odds ratios for these variables were not significantly different. Age and use of anticoagulants showed significantly high non-adjusted odds ratios for spontaneous eradication. A multivariate regression analysis revealed that only age showed significant correlation to the spontaneous eradication of *H. pylori* (Table 3).

The numbers of participants assigned to Groups A, B, C, and D in 2010 were 5613, 1059, 429, and 28, respectively. In contrast, the numbers of participants in Groups A, B, C, and D in 2015 were 5659, 1022, 402, and 46, respectively. Of the 7129 individuals analyzed, 23 had gastric cancer by December 2019. The cancer incidence rate among patients who underwent the ABC method in 2010 was 9/5613 (0.16%), 5/1059 (0.47%), and 9/457(1.97%) for Groups A, B, and C + D, respectively. The ABC method of 376 participants in 2010 and 2015 did not match because of discrepancies in the anti-*H. pylori* antibody titers and pepsinogen levels recorded in 2010 and 2015. When the participants whose ABC classifications in 2010 and 2015 did not match were classified into the higher-grade groups, Groups A, B, C, and D had 5543, 1036, 500, and 50 participants, respectively. The cancer incidence rate among participants who underwent the ABC method twice was 9/5543 (0.16%), 4/1036 (0.39%), and 10/550 (1.82%) for Groups A, B, and C + D, respectively. No significant difference was found in cancer incidence rates between participants who underwent the ABC method once and those who underwent it twice. In addition, the bootstrapping analysis results showed no significant differences in incidence rates (Table 4).

To exclude factors that could affect anti-*H. pylori* antibody titer or pepsinogen level, a subgroup analysis was performed after excluding cases of new *H. pylori* infection, spontaneous *H. pylori* eradication, history of upper gastrointestinal tract surgery, or PPI use. After excluding 11 participants with new infections, 20 that showed spontaneous *H. pylori* eradication, and 348 with a history of upper gastrointestinal tract surgery or PPI use, the remaining 6750 participants were included in the analysis. Of the 6750 participants, 18 had gastric cancer. The cancer incidence rates among participants who underwent the ABC method in 2010 were 7/5356 (0.13%), 4/990 (0.40%), and 7/404 (1.73%) for Groups A, B, and C + D, respectively. A total of 313 participants were classified into different groups based on whether they underwent the ABC method once or twice. When participants whose classifications in 2010 and 2015 did not match were classified into the higher grade groups, Groups A, B, C, and D had 5302, 957, 458, and 33 participants, respectively. The cancer incidence rates among participants who underwent the ABC method twice were 7/5302 (0.13%), 3/957 (0.94%), and 8/491 (1.63%) for Groups A, B, and C + D, respectively. No significant difference was found in the cancer incidence rate between participants who underwent the ABC method once and those who underwent it twice. The bootstrapping analysis results showed no significant differences in incidence rates as well (Table 5).

## 4. Discussion

In this study, we used large-scale health checkup data to determine whether it is useful to perform the ABC method twice and to examine the frequency of new *H. pylori* infections and spontaneous *H. pylori* eradication in adults. The results showed that there was no significant difference in the incidence rate of gastric cancer between the participants who underwent the ABC method once and those who underwent it twice. It appears that there is no need to repeat the ABC method in less than 5 years. In addition, 11 new *H. pylori* infections (11/7129 = 0.15%) and 20 spontaneous *H. pylori* eradication (20/7129 = 0.28%) were recorded within the 5-year study period. New *H. pylori* infection and spontaneous *H. pylori* eradication rates in adults were estimated to be 0.03% and 0.06% per person-year, respectively, in this study. These rates are lower than those reported in previous studies, where seroconversion (positive to a negative change in anti-*H. pylori* Immunoglobulin G (IgG) level and seroreversion (negative to a positive change in anti-*H. pylori* IgG level) rates were reported to be 0.53–0.70% and 0.11–24% per person-year, respectively [21,22,23]. In this study, new *H. pylori* infection was defined as an increase in anti-*H. pylori* antibody titer from <3 U/mL to ≥10 U/mL, whereas spontaneous *H. pylori* eradication was defined as a decrease in anti-*H. pylori* antibody titer from ≥10 U/mL to <3 U/mL. Compared with previous studies, cases in this study where anti-*H. pylori* antibody titers increased from 3–9.9 U/mL to ≥10 U/mL were uncategorized as new *H. pylori* infections. If new *H. pylori* infections in this study were defined as an increase in anti-*H. pylori* antibody titer from <10 U/mL to ≥10 U/mL as in previous studies, the rate of new *H. pylori* infection in this study would be 0.50% (36/7129). Similarly, if spontaneous *H. pylori* eradication was defined as a decrease in anti-*H. pylori* antibody titer from ≥10 U/mL to <10 U/mL, the rate of spontaneous *H. pylori* eradication in this study would be 1.3% (93/7129). Although these results were excluded from the main results, they are comparable to those of previous studies. In Japan, the prevalence of *H. pylori* has been decreasing because of improved sanitary conditions [24]. Recently, the prevalence of *H. pylori* in children aged 0–8 years was reported to be 1.9% without new *H. pylori* infections recorded during a 1-year follow-up period [25]. As the rate of *H. pylori* infection continues to decrease, the rate of new *H. pylori* infections is also expected to decline.

Miki et al. suggested that the ABC method can be used for national mass stratification for gastric cancer risk [11]. The effectiveness of the ABC method has been reported; however, the effectiveness of multiple ABC methods has not. In this study, we analyzed the incidence of gastric cancer at the end of 2019 in participants who underwent the ABC method in both 2010 and 2015. Even after excluding cases of new *H. pylori* infections, spontaneous *H. pylori* eradication, history of upper gastrointestinal tract surgery, and PPI use, no significant difference was found in cancer rates between participants who underwent the ABC method once and those who underwent it twice. These results were confirmed using a sufficiently large number of cases, and we considered them reasonable because they did not change after the bootstrapping analysis was performed.

Pepsinogen level is reported to barely fluctuate within approximately 10 years in more than 90% of adults [11]. It is possible that this stability in the pepsinogen level may be responsible for the lack of significant differences between participants who underwent the ABC method once and those who underwent it twice in Japan, where the rate of new *H. pylori* infection is low. Although assessment using the ABC method every 5 years did not improve gastric cancer risk stratification, it allowed for the identification of some cases of new *H. pylori* infections.

In this study, no participants with new *H. pylori* infection developed gastric cancer. The long-term risk of developing gastric cancer in adults with new *H. pylori* infections remains unknown. Therefore, further studies at long intervals are needed to clarify if the ABC method is sufficient once in a lifetime. The ABC method is used for stratifying the risk of gastric cancer in patients without a history of *H. pylori* eradication. As more people are eradicated of *H. pylori*, new screening methods for patients with a history of *H. pylori* eradication should be considered.

Spontaneous eradication of *H. pylori* was more likely to occur in the older adults and anticoagulant users in this study. Subjects with anticoagulants use were significantly older than those without anticoagulants use. The mean age of the 153 subjects with anticoagulants use was 57.7(±9.5) years, and the mean age of the 6976 subjects without anticoagulants use was 48.1 (±8.1) years. (*p* < 0.01) There is a correlation between age and anticoagulant use, and the multivariate regression analysis revealed that only age showed significant correlation to spontaneous eradication. The success rate of H. pylori eradication is higher in older adults than in the younger adults, and it has been suggested that decreased gastric acid capacity may be involved in this [26]. It is possible that the use of antibiotics for other reasons combined with the low gastric acidity may have caused an unintended eradication of *H. pylori*.

This study had some limitations. First, different reagents were used for measuring anti-*H. pylori* antibody titer in 2010 and 2015. Although we cannot exclude the possibility that the results may have been affected by this use of different reagents, we believe that the effect was minimal because of the high concordance rates of the anti-*H. pylori* antibody titers recorded. Second, definitions of new *H. pylori* infection and spontaneous *H. pylori* eradication were solely based on anti-*H. pylori* antibody titer, which may result in false-positive results. It has been reported that PGI and II are increased and PGI/PGII is decreased in patients with *H. pylori* infection, and that PGI and PGII are decreased and PGI/PGII is increased after the eradication of *H. pylori*. [27,28] In this study, the PG I and II levels of the 11 participants with new *H. pylori* infections tended to increase after infection; however, the difference in PG levels was not significant. In addition, the results indicated that PG I/PG II significantly decreased after infection (PG I: 71.7 ± 7.47 in 2010 and 78.7 ± 11.0 in 2015, *p* = 0.61; PG II: 10.3 ± 1.49 in 2010 and 21.6 ± 5.79 in 2015, *p* = 0.07; and PG I/PG II: 7.49 ± 0.83 in 2010 and 4.31 ± 0.30 in 2015, *p* < 0.01). In the 20 patients of spontaneous eradication, PGI decreased and PGI/PGII increased, although the differences were not significant; PGII decreased significantly (PG I: 34.8 ± 24.6 in 2010 and 32.5 ± 18.1 in 2015, *p* = 0.73; PG II: 13.2 ± 6.73 in 2010 and 7.95 ± 4.68 in 2015, *p* < 0.01; and PG I/PG II: 3.19 ± 2.51 in 2010 and 4.74 ± 2.68 in 2015, *p* = 0.06). These results suggest the usefulness of evaluating PG level and support the certainty of new *H. pylori* infections in this study. Third, not all participants were followed up until the end of 2019. In addition, EGD or upper gastrointestinal barium radiography is selected based on patient preference and gastric cancer screening methods are not standardized. However, 5222 of 7129 individuals (73.3%) were followed up until 2019 using upper gastrointestinal barium radiography or EGD. We believe that the number of participants who were followed up is sufficient to draw valid conclusions from this study’s results. Finally, we used a questionnaire to determine whether a participant had received *H. pylori* eradication therapy and whether it was successful. However, it should be noted that the evaluation of *H. pylori* eradication history in the general population using a questionnaire is considered reliable and valid [29].

In conclusion, this study demonstrated that performing the ABC method twice, 5 years apart, does not significantly improve gastric cancer risk stratification. In addition, this study showed that in Japan, the rates of new *H. pylori* infection and spontaneous *H. pylori* eradication over 5 years were 0.15% and 0.28%, respectively.

## Figures and Tables

**Figure 1 diagnostics-13-01284-f001:**
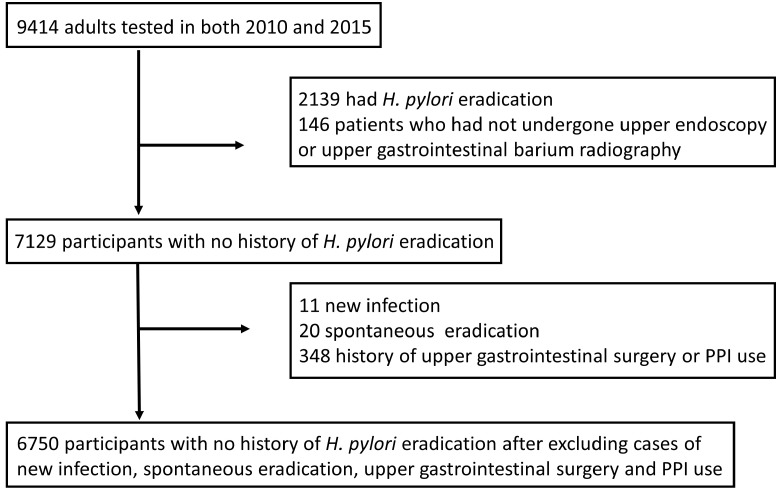
Study population selection flowchart. PPI, proton pump inhibitors; *H.pylori*, *Helicobacter pylori*. Of the 9414 participants, 7129 without a history of eradicating *H. pylori* were analyzed. Next, 6750 participants were analyzed, excluding those with a history of new *H.pylori* infection, spontaneous eradication, upper gastrointestinal surgery, or PPI use.

**Table 1 diagnostics-13-01284-t001:** Clinical characteristics of the study population.

				*p*-Value
Year		2010	2015	
Number		7129	7129	
Age		48.4 (±8.3)	53.4 (±8.3)	
Male		4145 (58.1%)	4145 (58.1%)	
BMI		23.1 (±3.4)	23.1 (±3.5)	0.23
Anti-*H. pylori* antibody titer, *n* (%)	<3 U/mL	5641 (79.1%)	5705 (80.0%)	
	3–9.9 U/mL	338 (4.7%)	331 (4.6%)	
	>10 U/mL	1150 (16.1%)	1093 (15.3%)	0.38
PG I		50.9 (±26.7)	60 (±36.8)	<0.01
PG II		9.9 (±7.1)	11.8 (±8.3)	<0.01
PG I/PG II		5.9 (±1.9)	5.7 (±1.8)	<0.01
ABC screening	Group A	5613	5659	
	Group B	1059	1022	
	Group C	429	402	
	Group D	28	46	0.11
Use of PPIs	Yes	97	217	
	No	7032	6912	<0.01
Upper gastrointestinal	Yes	59	54	
tract surgery	No	7070	7075	0.71
Smoking	Current	1470	1252	
	Former	1888	2132	
	Never	3771	3745	<0.01
Use of anticoagulants	Yes	153	242	
	No	6976	6887	<0.01
Use of steroids	Yes	57	46	
	No	7072	7083	0.28
Use of NSAIDs	Yes	449	579	
	No	6680	6550	<0.01

Groups A to D were classified using the ABC method, which is based on anti-*H. pylori* antibody titer and PG level. Abbreviations: BMI, body mass index; PG, pepsinogen; PPI, proton pump inhibitor; NSAIDs, non-steroidal anti-inflammatory drugs; *H. pylori*, *Helicobacter pylori.*

**Table 2 diagnostics-13-01284-t002:** Anti-*H. pylori* antibody titers recorded in 2010 and 2015.

		Anti-*H. pylori* Antibody Titers Recorded in 2010			
		<3 U/mL	3–9.9 U/mL	≥10 U/mL	
Anti-*H. pylori* antibody titers recorded in 2015	<3 U/mL	5577	108	20	5705
		78.20%	1.50%	0.30%	80.00%
	3–9.9 U/mL	53	205	73	331
		0.70%	2.90%	1.00%	4.60%
	≥10 U/mL	11	25	1057	1093
		0.20%	0.40%	14.80%	15.30%
		5641	338	1150	7129
		79.10%	4.70%	16.10%	100.00%

Abbreviation: *H. pylori*, *Helicobacter pylori.*

**Table 3 diagnostics-13-01284-t003:** Non-adjusted odds ratios for each variable recorded in 2010.

New *H. pylori* Infection	Non-Adjusted Odds Ratio (95% CI)	*p*-Value	Multivariate Odds Ratio (95% CI)	*p*-Value
Male vs. Female	3.0 (0.63–14.4)	0.17	2.1 (0.36–13.1)	0.37
Age *	1.03 (0.96–1.1))	0.41	1.01 (0.94–1.09)	0.66
BMI *	1.08 (0.94–1.25)	0.28	1.1 (0.83–1.20)	0.97
Upper intestinal tract surgery (yes vs. no)	NA	0.99	NA	
History of smoking (yes vs. no)	2.9 (0.89–9.7)	0.78	2.1 (0.46–10.0)	0.3
Use of PPIs (yes vs. no)	5.5 (0.60–51)	0.13	4.8 (1.02–6.2)	0.23
Use of anticoagulants (yes vs. no)	NA		NA	
Use of steroids (yes vs. no)	NA		NA	
Use of NSAIDs (yes vs. no)	1.5 (0.19–12)	0.70	2.4 (0.28–20.3)	0.46
Spontaneous eradication of *H. pylori*	Non-adjusted odds ratio (95% CI)	*p*-Value	Multivariate odds ratio (95% CI)	*p*-Value
Male vs. Female	2.4 (0.9–6.5)	0.09	1.7 (0.52–5.6)	0.36
Age *	1.13 (1.08–1.18)	<0.01	1.1 (1.05–1.16)	<0.01
BMI *	1.06 (0.95–1.19)	0.3	1.04 (0.90–1.19)	0.55
Upper intestinal tract surgery (yes vs. no)	2.2 (0.26–19)	0.47	1.8 (0.19–16.5)	0.62
History of smoking (yes vs. no)	1.2 (0.36–4.3)	0.73	1.1 (0.40–3.2)	0.78
Use of PPIs (yes vs. no)	NA		NA	
Use of anticoagulants (yes vs. no)	9.7 (3.2–30)	<0.01	3.2 (0.95–11.2)	0.08
Use of steroids (yes vs. no)	5.6 (0.7–45)	0.1	6.0 (0.72–50.9)	0.17
Use of NSAIDs (yes vs. no)	1.8 (0.4–8.3)	0.44	2.0 (0.44–9.4)	0.39

Abbreviations: BMI, body mass index; NA, not available; PPI, proton pump inhibitor; NSAIDs, non-steroidal anti-inflammatory drugs; CI, confidence interval; *H. pylori*, *Helicobacter pylori.* * Odds ratio per unit change.

**Table 4 diagnostics-13-01284-t004:** Comparison of cancer rates per year according to the ABC classification.

	Cancer Incidence Rate		*p*-Value			
	2010	2015	Fisher’s Exact Test	Bootstrapping Test	Difference between 2010 and 2015 (95% CI)	
Group A	9/5613	11/5659	0.82	0.33	−0.03%	(−0.13% to 0.13%)
	0.16%	0.19%				
Group B	5/1059	4/1022	1.00	0.36	0.08%	(−0.50% to 6.5%)
	0.47%	0.39%				
Group C + D	9/457	8/448	1.00	0.41	0.18%	(−1.4% to 2.0%)
	1.97%	1.79%				
	2010	ABC twice	Fisher’s exact test	Bootstrapping test	Difference between 2010 and ABC twice (95% CI)	
Group A	9/5613	9/5543	1.00	0.50	0.00%	(−0.16% to 0.16%)
	0.16%	0.16%				
Group B	5/1059	4/1036	1.00	0.36	0.09%	(−0.48% to 0.66%)
	0.47%	0.39%				
Group C + D	9/457	10/550	1.00	0.45	0.15%	(−1.6% to 1.9%)
	1.97%	1.82%				

Abbreviation: ABC twice: The ABC method performed twice; CI, confidence interval.

**Table 5 diagnostics-13-01284-t005:** Comparison of cancer rates per year according to the ABC method after excluding cases of new *H. pylori* infection, spontaneous *H. pylori* eradication, history of upper gastrointestinal tract surgery, and PPI use.

	Cancer Rate		*p*-Value			
	2010	2015	Fisher’s Exact Test	Bootstrapping Test	Difference between 2010 and 2015 (95% CI)	
Group A	7/5356	8/5391	0.80	0.35	−0.02%	(−0.18% to 0.11%)
	0.13%	0.15%				
Group B	4/990	2/956	0.69	0.19	0.19%	(−0.31% to 0.71%)
	0.40%	0.21%				
Group C + D	7/404	8/403	0.80	0.40	−0.25%	(−2.3% to 1.7%)
	1.73%	1.99%				
	2010	ABC twice	Fisher’s exact test	Bootstrapping test	Difference between 2010 and ABC twice (95% CI)	
Group A	7/5356	7/5302	1.00	0.43	−0.01%	(−0.14% to 0.11%)
	0.13%	0.13%				
Group B	4/990	3/957	1.00	0.38	0.09%	(−0.42% to 0.60%)
	0.40%	0.31%				
Group C + D	7/404	8/491	1.00	0.47	0.10%	(−1.5% to 1.8%)
	1.73%	1.63%				

Abbreviation: ABC twice; The ABC method performed twice; CI, confidence interval; PPI, proton pump inhibitor; *H. pylori*, *Helicobacter pylori.*

## Data Availability

The data are not publicly available due to the decision of the Ethics Review Committee.

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
