# Peer review of "Performing the ABC Method Twice for Gastric Cancer Risk Stratification: A Retrospective Study Based on Data from a Large-Scale Screening Facility"

_diagnostics, 2023, doi:10.3390/diagnostics13071284_

Round 1

Reviewer 1 Report

General comments:

 This study is an examination of patients who had undergone screening by ABC method twice to evaluate whether there is a change in the risk stratification ability for gastric cancer by the multiple examinations.

Although not the main objective of the study, it presents very important and interesting data, such as the proportion of newly infected HP patients and spontaneously eradicated HP patients over a 5-year period, and evaluates HP infection status based on the results of HP antibody tests, using stricter criteria than in previous reports. This study could provide more accurate data about the natural course of HP infection status in adults. The study presents important data and is novel in that it presents more accurate data.

However, regarding the main conclusion of the study, it may be difficult to interpret. Though there was no significant change in risk stratification ability for gastric cancer by the multiple screening tests by ABC method, it does not mean that the risk stratification ability of a single exam and multiple exams are equal, necessarily. Equivalency was not evaluated.

Also, the message and usefulness of this data for clinical practice is unclear. Clinical suggestions led by this study should be described concretely.
As for the manuscript, it is very well written, logical, easy to read, and properly examined, including statistical analysis.

The paper provides very interesting data but it still needs considerable revision to be acceptable for the journal “diagnostics”. I apologize for being so harsh in my review, but hopefully, my comments will help you to improve your manuscript and get it published.

Major comments

#1 What is the definition of spontaneous eradication?

Please give a clear definition of the term “spontaneous eradication”.
I think that HP previous infection cases other than cases eradicated by drug administration are generally classified into two groups; accidental eradication and natural eradication. To which group does the “spontaneous eradication” belong?  This is an important point, so please provide accurate definitions in the material and methods section.

#2 Diagnosis of new HP infection and spontaneous eradication cases by HP antibodies
It is valuable to provide the data on the new HP infection and spontaneous eradication over 5 years in adults based on stricter criteria, which is different from the previous report as authors mentioned in the discussion section (lines 251-259). The authors defined new HP infection in adults as an increase in HP antibody value from less than 3 to 10 or more, and spontaneous eradication as a decrease in HP antibody value from 10 or more to less than 3. However, strictly speaking, it is insufficient to diagnose a new infection or spontaneous eradication strictly by HP antibodies alone; it is a limitation. For accurate diagnosis of HP infection status, endoscopic examination is essential and not only HP antibody value but also PG value should be taken into consideration.

They should discuss the validity of their criteria showing the detailed data of new HP infection cases and spontaneous eradication cases (PG value, atrophy in endoscopy/barium radiography, group according to ABC method, and so on) in this study.

Please also show from which group to which group the classification changed in new HP infection cases and spontaneous eradication cases.

#3 Group change according to ABC screening criteria

Analyzing the group change by 2nd ABC screening test is the main subject of this study. Detailed description is desirable as this is the core data. Please show the number of cases that changed, A to B, A to C, A to D, B to A, B to C, B to D, C to A, C to B, C to D, D to A, D to B, and D to C in Table 4 or Table 5.

Was there any unreasonable change in the group? Please provide the characteristics of the cases in which group classification changed. By examining these factors, it would be possible to consider what factors, other than new infection or spontaneous eradication, may have caused the change in classification based on the ABC method. The group change may be partially due to the issue of the accuracy of the examination.
For example, the increase of Group A cases by a large amount in the second exam shown in Tables 1, 4, and 5 seems rather unreasonable. I can understand that there are cases in which atrophy becomes apparent. Thus, the change from group A to the other group is reasonable. But, is it possible that gastritis disappears and the group changes from other groups to group A, other than spontaneous eradication?

#4 Is the ABC screening test commonly recommend to be performed only once in life in practice? Is it often conducted repeatedly? Please describe how it is operated in practice in the introduction section. Please provide more detailed information on why the authors aimed to explore the significance of the 2nd ABC screening test in this study. Furthermore, based on the results of this study, how many times ABC screening does the author suggest should be conducted in future practice? Please describe the authors’ opinion on this point in your discussion.

#5 The third limitation in the discussion section (lines 306-309)

As the authors described as the third limitation, only 73% of patients are followed. Furthermore, the surveillance methods (EGD or barium radiography), and the follow-up intervals are not standardized, this is also problematic.
For monitoring carcinogenesis and evaluating the risk stratification ability of screening tests, it would be desirable to evaluate a group of patients with a unified follow-up method (eg. annual EGD for 5 years). These are the huge limitations of this study.

Minor comments

#6 Lines 188-189; “Age and use of anticoagulants showed significantly high non-adjusted odds ratios for spontaneous eradication”

Why is spontaneous eradication observed more in elderly patients and anticoagulant users? Is there any speculation?

#7 Lines 275-276; “Although these results were confirmed using a sufficiently large number of cases, we considered them robust because they did not change after bootstrapping analysis was performed.”

The meaning of the sentence above is a bit unclear. Certainly, the number of patients who underwent ABC screening is sufficient, but the number of cases who developed gastric carcinoma is extremely small. Though there was no significant difference in cancer risk stratification accuracy between the multiple ABC group and single ACC group, it may be just due to the smallness of cancer cases.

The bootstrapping analysis is an effective method in analysis for a population where normality cannot be confirmed. But the multiple-time bootstrapping analysis does not compensate for the smallness of the number in statistical analysis. It seems an overstatement to say that the results are robust just because they did not change after bootstrapping analysis. 

Author Response

We thank the reviewer #1 reading and commenting on our manuscript.

#1 What is the definition of spontaneous eradication?
Please give a clear definition of the term “spontaneous eradication”.
I think that HP previous infection cases other than cases eradicated by drug administration are generally classified into two groups; accidental eradication and natural eradication. To which group does the “spontaneous eradication” belong? This is an important point, so please provide accurate definitions in the material and methods section.

We appreciate this very informative comment. It is difficult to precisely distinguish between "accidental eradication" and "natural eradication. In this study, "spontaneous eradication" was defined as anything other than the intended use of H. pylori eradication medication. For greater clarity, we have modified it as shown line 119-124.

#2 Diagnosis of new HP infection and spontaneous eradication cases by HP antibodies It is valuable to provide the data on the new HP infection and spontaneous eradication over 5 years in adults based on stricter criteria, which is different from the previous report as authors mentioned in the discussion section (lines 251-259). The authors defined new HP infection in adults as an increase in HP antibody value from less than 3 to 10 or more, and spontaneous eradication as a decrease in HP antibody value from 10 or more to less than 3. However, strictly speaking, it is insufficient to diagnose a new infection or spontaneous eradication strictly by HP antibodies alone; it is a limitation. For accurate diagnosis of HP infection status, endoscopic examination is essential and not only HP antibody value but also PG value should be taken into consideration.

They should discuss the validity of their criteria showing the detailed data of new HP infection cases and spontaneous eradication cases (PG value, atrophy in endoscopy/barium radiography, group according to ABC method, and so on) in this study. Please also show from which group to which group the classification changed in new HP infection cases and spontaneous eradication cases.

As the reviewer pointed out, there is a limitation to discussing the status of H. pylori infection based on H. pylori antibodies alone. In order to be more robust about the status of H. pylori infection, PG values were also added. (line312-314,319-323 ) The endoscopic findings were retrospectively reviewed, but many lacked gastric mucosal changes and were not described in this study. This is because the degree of atrophy is small in the early stages of new H. pylori infection, making it difficult to classify H. pylori infection status by endoscopic findings. Similarly, for spontaneous eradication, it was difficult to mention the status of H. pylori based on endoscopic findings alone.

In the 11 patients in the newly infected with H. pylori group, all were in group A in 2010, but all were in group B in 2015.

In the 20 patients in the spontaneous eradication group, there were 9 in group B and 11 in group C in 2010, and 14 in group A and 6 in group D in 2015. As discussed in lines 319-323, this is because there have been cases in which spontaneous eradication of H. pylori has resulted in an increase in PGI/PGII and the pepsinogen test has gone from positive to negative.

#3 Group change according to ABC screening criteria
Analyzing the group change by 2nd ABC screening test is the main subject of this study. Detailed description is desirable as this is the core data. Please show the number of cases that changed, A to B, A to C, A to D, B to A, B to C, B to D, C to A, C to B, C to D, D to A, D to B, and D to C in Table 4 or Table 5.
Was there any unreasonable change in the group? Please provide the characteristics of the cases in which group classification changed. By examining these factors, it would be possible to consider what factors, other than new infection or spontaneous eradication, may have caused the change in classification based on the ABC method. The group change may be partially due to the issue of the accuracy of the examination.
For example, the increase of Group A cases by a large amount in the second exam shown in Tables 1, 4, and 5 seems rather unreasonable. I can understand that there are cases in which atrophy becomes apparent. Thus, the change from group A to the other group is reasonable. But, is it possible that gastritis disappears and the group changes from other groups to group A, other than spontaneous eradication?

The ABC method classification for 2010 and 2015 was presented in the image below. As for the reason for the increase in Group A in 2015, there is the impact of the H. pylori antibody titer mismatch as well as spontaneous eradication.

For example, 103 were classified from group B in 2010 to group A in 2015.Eight of them were in the spontaneous eradication group that changed from H. pylori antibody titers of 10 or more to less than 3. The remaining 95 patients were in the "negative high" group with H. pylori antibody levels ranging from 3-10. In the ABC classification, H. pylori antibody 3 or higher is considered positive. On the other hand, in our study, in order to reduce false positives, those with H. pylori antibodies that changed from 10 or more to less than 3 were considered spontaneous eradication, and those with 3-9.9 were excluded. Due to the strict definition of spontaneous eradication, there appears to be a lot of change from group B to group A other than spontaneous eradication. H. pylori antibody titers for 2010 and 2015 are presented (table2).

#4 Is the ABC screening test commonly recommend to be performed only once in life in practice? Is it often conducted repeatedly? Please describe how it is operated in practice in the introduction section. Please provide more detailed information on why the authors aimed to explore the significance of the 2nd ABC screening test in this study. Furthermore, based on the results of this study, how many times ABC screening does the author suggest should be conducted in future practice? Please describe the authors’ opinion on this point in your discussion.

In Japan, it is believed that ABC screening should be performed once in a lifetime, but in practice, many people undergo ABC screening repeatedly.
It is difficult to draw conclusions from this study alone, but at this point it does not appear necessary to repeat ABC screening in less than 5 years.
We rewrote some sentences in introduction and added our opinion in discussion. (line 64-68, line 245-246,290-291)

#5 The third limitation in the discussion section (lines 306-309)
As the authors described as the third limitation, only 73% of patients are followed. Furthermore, the surveillance methods (EGD or barium radiography), and the follow-up intervals are not standardized, this is also problematic.
For monitoring carcinogenesis and evaluating the risk stratification ability of screening tests, it would be desirable to evaluate a group of patients with a unified follow-up method (eg. annual EGD for 5 years). These are the huge limitations of this study.

As the reviewer pointed out, surveillance rates and the lack of uniformity in surveillance methods are limitations. We have made a correction to the discussion section to state this in a clear manner.(line 325-327)

Minor comments
#6 Lines 188-189; “Age and use of anticoagulants showed significantly high non-adjusted odds ratios for spontaneous eradication”
Why is spontaneous eradication observed more in elderly patients and anticoagulant users? Is there any speculation?

We appreciate reviewer’s instructive comment. Although it cannot be said with certainty due to the small number of cases, it has been reported that the eradication rate is higher in older age groups. In this study, spontaneous eradication tended to occur in older subjects, and since anticoagulants use was more common in the older subjects. The mean age of the 153 subjects with anticoagulants use was 57.7(±9.5) years and the mean age of the 6976 subjects without anticoagulants use was 48.1 (±8.1) years. (p<0.01) There is correlation between age and anticoagulant use, and multivariate regression analysis revealed that only age showed significant correlation to spontaneous eradication. We rewrote some sentences in line295-305.

#7 Lines 275-276; “Although these results were confirmed using a sufficiently large number of cases, we considered them robust because they did not change after bootstrapping analysis was performed.”
The meaning of the sentence above is a bit unclear. Certainly, the number of patients who underwent ABC screening is sufficient, but the number of cases who developed gastric carcinoma is extremely small. Though there was no significant difference in cancer risk stratification accuracy between the multiple ABC group and single ACC group, it may be just due to the smallness of cancer cases.
The bootstrapping analysis is an effective method in analysis for a population where normality cannot be confirmed. But the multiple-time bootstrapping analysis does not compensate for the smallness of the number in statistical analysis. It seems an overstatement to say that the results are robust just because they did not change after bootstrapping analysis.

As the reviewer pointed out, stating "robust" after the boost trap analysis seems to be an overstatement, so we have corrected the wording. line278-280

Reviewer 2 Report

Overall:

 The authors investigated whether the repetition of the ABC method is beneficial for gastric cancer risk stratification and examined the occurrence rates of new H. pylori infections and spontaneous H. pylori eradication in Japan using recent large-scale health checkup data. The study's findings suggest that performing the ABC method twice, with a 5-year interval, does not significantly enhance gastric cancer risk stratification.

 Comments:

1.       Could you provide information on how the participants who underwent the ABC method attained H. pylori eradication? Furthermore, why were 1150 participants not eradicated, and is there any inherent bias in this group?

2.       How were the participants instructed to undergo endoscopy or upper gastrointestinal barium radiography based on the ABC method results?

3.       Is it appropriate to compare the cancer incidence rate between the 2010 and 2015 groups, which had different observation periods? Was the cancer incidence rate based on the number of gastric cancer cases that arose between 2015 and 2019? Were gastric cancer cases that occurred between 2010 and 2015 excluded because of endoscopic or surgical resection?

4.       Please elaborate on the greater frequency of spontaneous eradication in cases involving anticoagulant use.

5.       Figure 1:  6750 participants…., upper gastrointestinal and PPI use→  6750 participants…., upper gastrointestinal surgery and PPI use

Author Response

We thank the reviewer #2 for reading and commenting our manuscript.

The authors investigated whether the repetition of the ABC method is beneficial for gastric cancer risk stratification and examined the occurrence rates of new H. pylori infections and spontaneous H. pylori eradication in Japan using recent large-scale health checkup data. The study's findings suggest that performing the ABC method twice, with a 5-year interval, does not significantly enhance gastric cancer risk stratification.

Comments:
1. Could you provide information on how the participants who underwent the ABC method attained H. pylori eradication? Furthermore, why were 1150 participants not eradicated, and is there any inherent bias in this group?

In Japan, eradication of H. pylori became covered by national health insurance in 2013. Since then, eradication of H. pylori has been recommended for patients who test positive for H. pylori antibodies during medical examinations. It is up to each individual to decide whether or not to follow the recommendation and perform H. pylori eradication.
Since this study compares the results of measurements in 2010 and 2015 of exactly the same participants, there is no bias from this perspective.

2. How were the participants instructed to undergo endoscopy or upper gastrointestinal barium radiography based on the ABC method results?

Although the ABC method can clearly stratify the risk of gastric cancer, it has not yet been officially recognized as a screening method in Japan. Therefore, annual screening by EGD was recommend for the participants in groups B, C, or D, but the actual choice of EGD or upper gastrointestinal barium radiography was left to the individual regardless of the results of the ABC method. This is one of the limitations of this study, we rewrote a sentences in discussion in line 325-327.

3. Is it appropriate to compare the cancer incidence rate between the 2010 and 2015 groups, which had different observation periods? Was the cancer incidence rate based on the number of gastric cancer cases that arose between 2015 and 2019? Were gastric cancer cases that occurred between 2010 and 2015 excluded because of endoscopic or surgical resection?

We thank the reviewer this informative comment.
The gastric cancer incidence rates are those that occurred between 2010 and 2019 for all groups. Among 7129 patients, 23 gastric cancer carcinogenesis that occurred between 2010 and 2019 were classified based on the ABC method in 2010 and 2015.(Table 4) For this study, we did not distinguish between gastric cancers that occurred before 2015 and those that occurred after, since it is impossible to compare exactly the same subjects in 2010 and 2015 if we excluded those who developed gastric cancer before 2015.
5 cases of gastric cancer occurred between 2010 and 2015, and 18 cases occurred after 2015.
The five gastric cancer cases that developed between 2010 and 2015 were two Group A, two Group B, and one Group C case in 2010, and four Group A and one Group B case in 2015.
Of the 6750 patients after excluding new H. pylori infection and other factors, 18 gastric cancers that occurred between 2010 and 2019 were classified based on the ABC method in 2010 and 2015.(Table 5)
3 cases of gastric cancer occurred between 2010 and 2015, and 15 cases occurred after 2015.
The three gastric cancer cases that developed between 2010 and 2015 were all Group A cases in 2010 and two Group A and one Group B case in 2015.

4. Please elaborate on the greater frequency of spontaneous eradication in cases involving anticoagulant use.

We appreciate reviewer’s instructive comment. Although it cannot be said with certainty due to the small number of cases, it has been reported that the eradication rate is higher in older age groups. In this study, spontaneous eradication tended to occur in older subjects, and since anticoagulants use was more common in the older subjects, there may be possibility of this association. In fact, Multivariate odds adjustment showed only age significantly correlated with spontaneous eradication. Thanks to the reviewer’s comment, we added results of multivariate analysis to Table3. We also rewrote in line 188-190 and line 295-305, as follows.
Multivariate regression analysis revealed that only age showed significant correlation to spontaneous eradication of H. pylori (Table 3).

Spontaneous eradication of H. pylori was more likely to occur in the older adults and anticoagulant users in this study. Subjects with anticoagulants use were significantly older than those without anticoagulants use. The mean age of the 153 subjects with anticoagulants use was 57.7(±9.5) years and the mean age of the 6976 subjects without anticoagulants use was 48.1 (±8.1) years. (p<0.01) There is correlation between age and anticoagulant use, and multivariate regression analysis revealed that only age showed significant correlation to spontaneous eradication. The success rate of H. pylori eradication is higher in older adults than in the younger adults, and it has been suggested that decreased gastric acid capacity may be involved in this [26]. It is possible that the use of antibiotics for other reasons combined with the low gastric acidity may have caused unintended eradication of H. pylori.

5. Figure 1: 6750 participants……., upper gastrointestinal and PPI use 6750 participants……., upper gastrointestinal surgery and PPI use

We thank the reviewer for pointing this out. We have corrected it as indicated. (Figure1)

Reviewer 3 Report

1) Pepsinogen level barely fluctuate within approximately 10 years, here you performed ABC method twice 5 years apart, this might   be the cause for no difference in gastric cancer risk. This may not be true when you perform ABC at longer intervals. So, why you didn't plan ABC at more than 10 years or longer intervals in patients without history of H Pylori eradication?

2) How you obtained written consent from each participant in this retrospective data base study?

Author Response

We thank the reviewer #3 for reading and commenting our manuscript.

1) Pepsinogen level barely fluctuate within approximately 10 years, here you performed ABC method twice 5 years apart, this might be the cause for no difference in gastric cancer risk. This may not be true when you perform ABC at longer intervals. So, why you didn't plan ABC at more than 10 years or longer intervals in patients without history of H Pylori eradication?

As the reviewer commented, a five-year interval is short and a 10-year interval could have yielded different results. However, there is not enough information on serum pepsinogen and anti- H. pylori antibody in 2020 or later, so we examined the data at 5 years in this study. Risk assessment at long-term intervals is a subject for our future research.

2) How you obtained written consent from each participant in this retrospective data base study?

A prospective observational study was conducted starting in 2010 (UMIN 000013761), and written consent forms were obtained from all participants at the time of its entry. In addition, an opt-out was used for this study.

Round 2

Reviewer 1 Report

The authors responded and addressed all my comments appropriately. The manuscript has much improved and become easier to understand. The additional data and descriptions are detailed and helpful. I appreciate the sincere response by the authors.